# OpenReview forum: "Dynamic Incremental Code Embeddings (DICE):  A Real-Time Communication Protocol for Multi-Agent Reinforcement Learning"
_ICLR.cc/2026/Conference — Submitted to ICLR 2026_

### Official Review · Reviewer_FmUF · 2025-10-20

**Soundness:** 2
**Presentation:** 2
**Contribution:** 2
**Rating:** 4
**Confidence:** 4

**Summary:**

This work proposes Dynamic Incremental Code Embeddings (DICE) for MARL-based collaborative code completion. The authors claim that it is a real-time communication protocol to address the inefficiency of static or periodically updated embeddings. It consists of two major components: dynamic semantic drift encoding (DSDE) and dynamic contextual embedding adaptation (DCEA). Through experiments, the authors show that their method performs better than three other methods in terms of Suggestion Quality, Computational Efficiency, and Semantic Coherence.

**Strengths:**

The advantages of this work are listed as follows.
1. MARL-based code completion is an important topic.
2. It is interesting to see the application of MARL to code completion.

**Weaknesses:**

The disadvantages of this work are listed as follows.
1. Unclear motivation, it does not propose a new MARL algorithm or a new MARL-communication mechanism. It proposed a new embedding method.
2. Incomplete related work regarding MARL, there are quite a lot of MARL algorithms that are missing. For example, QMIX, QTran, ResQ, MAPPO are popular MARL methods.
3. Incomplete related work regarding code completion. For example, ChatDev is a famous multi-agent-based programming framework.
4. Limited novelty, the proposed embedding method seems trivial without much insight. It is OK to use simple formulations or simple methods; the insights are required by the ICLR community.

Minors:
1. lines 067-068, outdated reference, and existing MARL communication does use dynamic communication protocols, for example, DGL is a GAT-based MARL communication method.
2. lines 093-094, wrong word2vec citation.
3. lines 259-260, What are the action spaces?

**Questions:**

What research gap does the proposed embedding method fill? How does it enhance existing MARL for code completion? Comparisons with MARL algorithms (QMIX, MAPPO) and code frameworks (ChatDev) are needed. Also please explain their differences and your advantages.
Provide insights behind the embedding method and evidence to prove its non-triviality.

---

### Official Review · Reviewer_mVVJ · 2025-10-24

**Soundness:** 2
**Presentation:** 1
**Contribution:** 2
**Rating:** 2
**Confidence:** 4

**Summary:**

The work formalizes the problem of multiple agents collaborating to complete source code as a multi-agent reinforcement learning problem. Motivated by computational cost of existing approaches and continual changes of the context of the agents, i.e. changes in the code, they propose a novel approach with two key components: (1) dynamic semantic drift encoding (DSDE) that allows to efficiently update context of agents, and (2) dynamic contextual embedding adaptation (DCEA) that incorporates communication between multiple agents to update their context. The approach is evaluated in a benchmark derived from the CodeSearchNet dataset. Embeddings of code are obtained using fine-tuned GPT 3.5-turbo and CodeBERT, and the approach is compared to baselines using static embeddings, periodically re-training agents, and a centralized MARL baseline. The proposed approach is found to lead to higher match with the ground-truth edits in the dataset, significantly reduce cost, and leads to less redundant changes between agents.

**Strengths:**

The problem formulation of the work is original and interesting. While prior exists, and is being cited within the work, that uses a similar formulation, this work further adds further considerations such as how to reduce the cost of this task across multiple agents, and how to improve the effectiveness of the inter-agent collaboration through communication.

The proposed DICE approach naturally builds on the outlined challenges of existing solutions and is conceptually simple which I consider a benefit. I also appreciate that the authors have considered several aspects throughout their evaluation, such as the computational cost and redundancy between agents that add further nuance to the results.

Unfortunately, the work also suffers from a lack of clarity and depth in motivation, methodology, and experimentation, that are further discussed below.

**Weaknesses:**

In its current form, I am afraid that the work is not of the quality I would expect to justify acceptance at a venue like ICLR. Below, I try to give concrete weaknesses and highlight any weaknesses that I see as critical / major with (**Major**). I'd expect these to be addressed for this work to be considered for acceptance.
### Motivation & Problem Setting
1. **Major**: Given the focus of the work to frame multi-agent code completion as a MARL problem, I would have expected to see a section that defines the Dec-POMDP, Markov game/ (partially observable) stochastic game but such a definition of the problem is absent. Section 3.3 only loosely describes the multi-agent problem. Latter parts of the work, see e.g. Equation (6, 7), appear to refer to parts of the environment such as rewards, states, and actions but it was never defined what these exactly are. I believe that a clear definition of the problem would significantly help to substantiate any claims made.
2. **Major:** The work appears to start from the premise that code completion should be solved collaboratively by multiple agents but it is not clear why exactly that is the case. What are the unique benefits of framing the problem as a multi-agent problem as opposed to a single-agent decision-making problem? While I can think of potential reasons, there are also challenges within that setting that are insufficiently discussed.
3. When motivating their work, the authors state  "Existing MARL frameworks [...] lack mechanisms to handle the continuous updates and semantic coherence required for collaborative coding" (Section 1) -- it is not clear to me why that is true and what exactly is meant with "semantic coherence". No citation or evidence is provided for the statement.
4. The authors often use terms such as "semantic coherence", "static embeddings" (embeddings of what?), "periodic recomputations" (recomputation of what?) that are not clear to me within the given context. Given these terms appear central to the paper narrative, I would highly suggest to clearly define/ introduce them.
### Methodology
5. **Major:** The authors claim the following about their approach of computing embeddings: "This formulation guarantees that the computational cost is proportional to the magnitude of edits and not to the size of the codebase." (Section 4.1). This appears to be a central claim to the work but it is not clear to me why that is true, and why simple alternatives could not remedy inefficiencies claimed in existing approaches. See the following questions
	1. Would the embedding size $e_t$ not scale with the size of the codebase in which case the input of the network scales with the codebase?
	2. Why is the computational cost of Eq (1) proportional to the edit difference $\Delta C_t$?
	3. This claim is made in contrast to existing approaches that use static embeddings and the authors explicitly refer to word2vec-style models earlier (Section 2.1) as an example of such approaches. Wouldn't it possible to use word2vec embeddings and only re-compute these embeddings for parts of the code that have been changed? In this case, the cost of such a simple approach would not scale with the overall codebase size but with the magnitude of edits captured by $\Delta C_t$, similar to the claim made about the proposed approach of computing embeddings.
6. **Major:** Several aspects of the proposed approach outlined in Section 4 are unclear to me
	1. DSDE formulation in Eq (2) and (3) are written with time-based subscripts and no agent indices while DCEA Eq (4-7) denote further formulations with agent subscripts. This discrepancy is somewhat confusing. Is the DSDE model shared across all agents and all agents use Eq (2-3) to compute their initial embeddings, or are these agent-specific?
	2. Eq (2) mentions a hidden state that summarizes historical context but it is not defined where this input for the MLP network is obtained from, and how it is updated. Is this term agent specific?
	3. Eq (4) contains a $u^T$ term but this vector is undefined.
	4. Eq (4) and the description for DCEA discusses messages of agents that are being used to compute embeddings but it is never defined how agents generate their messages.
	5. Eq (5) LHS appears to define the output of DCEA as $e^{t+1}_i$ while Eq (4) describes the output of the model as $\tilde{e}^t_i$. Are these meant to be identical, and if not, how are they different?
### Empirical Evaluation
7. **Major:** It is unclear to me what represents the evaluation environment. Section 5.1 states that the benchmark is derived from the CodeSearchNet dataset but it is unclear to me how this translates to an online learning environment. Below are some more concrete questions related to this evaluation task:
	1. Given the problem formulation is mentioned to be MARL, I would expect a test environment to specify state space, action space, rewards but all of these are absent. This connects to my major concern about the absence of any problem formalism (see Weakness 1.).
	2. The authors state that "we can replay the coding sessions" which suggests that algorithms are trained purely offline from the edit history dataset? Is that correct? Does this still represent a MARL environment?
8. I have concerns about the baselines chosen for this work and whether they are sufficient.
	1. **Major:**  Connected to weakness 5.3, it is unclear to me why it is not possible to obtain embeddings (e.g. word2vec-based) that can be efficiently re-computed for edited parts of the codebase only. This would appear to enable efficient updates of embeddings without requiring re-computation of embeddings across the entire codebase or fine-tuning as done in the SE and PR baselines.
	2. This algorithm is stated to build on top of MADDPG (as per citation). That algorithm uses centralized training but decentralized execution while the baseline is referred to as centralized which might imply fully centralized. Would the authors be able to clarify whether the policies of the centralized MARL algorithm allow for decentralized execution or are fully centralized? This again connects back to the missing assumptions in the problem setting (see Weakness 1).
9. **Major:** For any collaborative MARL problem, I would have expected to see some learning curves that show evaluation returns throughout training. Instead, the primary metric of quality of the generated code appears to be accuracy as a measure of distance between the ground truth changes in the dataset and the agents. This metric appears to be assuming a supervised learning type of problem where the goal is to imitate the human edits. Why is that a good metric for coding agents? It appears that coding agents could do very different edits compared to humans that perfectly solve the coding problems being tackled -- I would have not considered those poor solutions.
10. Section 5.1 states that "all experiments ran [...] with 5 agents per team" -- how many teams were there in the problem task? And how do teams differ between each other? Does each team edit different parts of the codebase? This is never discussed and, again, falls back to a missing problem formulation (Weakness 1).

**Questions:**

1. What are the unique benefits of framing the problem of code completion as a multi-agent problem as opposed to a single-agent decision-making problem?
2. Why does the computational cost of the proposed DICE approach not scale with the codebase size but be proportional to magnitude of edits, as claimed in Section 4.1? Would $e_t$ as input to the system not scale with the size of the codebase?
3. Is the DSDE model shared across all agents and all agents use Eq (2-3) to compute their initial embeddings, or are these agent-specific?
4. How do agents compute their messages used in Eq. (4)?
5. What are the states, actions, and rewards within the benchmark environment? Are algorithms trained purely offline from a dataset, or is there any online learning that takes place within the environment?
6. Word2vec-based embeddings are mentioned as an example for static embeddings that are inefficient. Wouldn't it be possible to compute these embeddings in a way that only requires re-computation of edited parts of the codebase rather than re-computing embeddings across the entire codebase, as claimed? If you claim this is not possible, why is it not possible?
7. Why do you think that distance to "ground-truth" human edits in your evaluation is a good metric of quality of the generated code? Would it not be possible for coding agents to write very different code to humans that still perfectly solves the problem while another solution might be very similar to the human code but have key errors?

---

### Official Review · Reviewer_qi8J · 2025-10-25

**Soundness:** 3
**Presentation:** 2
**Contribution:** 2
**Rating:** 4
**Confidence:** 4

**Summary:**

The paper proposes the DICE framework, which models code embeddings as a shared, incrementally evolving state. It combines DSDE, a continuous-time semantic drift encoder, with DCEA, a GAT-based module for cross-agent contextual adaptation. In multi-agent collaborative code completion experiments on CodeSearchNet, DICE outperforms static and periodically retrained baselines.

**Strengths:**

1.	The problem is clearly defined — the paper directly targets the limitation of static embeddings in collaborative code completion by proposing a dynamic, incremental embedding approach.

2.	The overall framework is well-structured and conceptually coherent, with a clear system design.

**Weaknesses:**

1. The design of DSDE and DCEA closely resembles existing communication-based MARL works built on graph neural networks (e.g. GraphComm, CommNet), and the overall novelty is limited.

2. Although this paper claims to provide theoretical analysis of convergence and scalability, there is no explicit derivation, theorem, or complexity proof in the main text. The stability boundaries of DSDE and the asynchronous consistency of DCEA are not formally addressed, leaving the theoretical support relatively weak.

**Questions:**

1.In Section 4.2 , DCEA is described as a GAT-based dynamic contextual adaptation, but similar mechanisms exist in prior MARL works such as CommNet, DGN, and GraphComm. Could the authors clarify the key improvements of DCEA and whether it offers new stability or generalization under asynchronous or delayed communication?

2.The abstract and Section 5.3 claim theoretical analysis of convergence and scalability, but no formal theorems or proofs are provided. Could the authors present concrete evidence for the “linear scalability” claim and clarify whether it refers to computational, communication, or sample complexity?


3.Section 4.2 states that “DSDE captures semantic drift with a continuous-time process,” followed by the update equation
$\Delta s_t = \mathrm{MLP}_{\theta_d}([h_{t-1}; \Delta C_t])$
and a tanh activation. Is this continuous-time process modeled via an ordinary differential equation (ODE), a time-encoding function, or a discrete-step approximation?

4.Experiments are based on the modified CodeSearchNet dataset. Do the author evaluate DCEA’s scalability in large-scale MARL or considered hierarchical or sparse communication to reduce overhead?

If the authors address my concerns, I will increase my score.

references:

[1] Graphcomm: A Graph Neural Network Based Method for Multi-Agent Reinforcement Learning
[2] Multi-Agent Graph-Attention Communication and Teaming

---

### Meta-Review · Area_Chair_pX4o · 2025-12-19

**Summary:**

This work proposes Dynamic Incremental Code Embeddings (DICE) for MARL-based collaborative code completion.  It consists of two major components: dynamic semantic drift encoding (DSDE) and dynamic contextual embedding adaptation (DCEA). Through experiments, the authors show that their method is promising.

The strengths of this work are listed as follows.
1. The topic is important and timely.
2. The overall framework is well-structured and conceptually coherent.

The weaknesses of this work are listed as follows.
1. The novelty of this work is limited. It is similar to several existing works as it is pointed out by some reviewers.
2. Incomplete related work.

**Reviewer Concerns:**

The authors did not respond to the reviews.

**Reviewer Scores:**

The authors did not respond to the reviews.

---

### Decision · Program_Chairs · 2026-01-26

Reject